TOPICAL REVIEW

# A tight squeeze: how do we make sense of small changes in microvascular diameter?

Harvey Davis and David Attwell 

*Department of Neuroscience, Physiology & Pharmacology, University College London, London, UK*

The peer review history is available in the Supporting Information section of this article (https://doi.org/10.1113/JP284207#support-information-section).

Diameter — Arteriole

Pericytes and vascular smooth muscle cells are both major contributors to blood flow control

Capillary bed

Arteriole

Capillary

Capillaries have a smaller diameter than arterioles, which drastically increases the effect of pericyte contraction upon vascular resistance

Timing

Capillaries dilate before arterioles thereby providing the temporally first reduction in vascular resistance.

Occlusions

Flow

Pericyte mediated capillary constriction increases the probability of capillary occlusion by neutrophils and other blood cells

The Journal of **Physiology**

**Abstract**  The brain is an energetically demanding tissue which, to function adequately, requires constant fine tuning of its supporting blood flow, and hence energy supply. Whilst blood flow was traditionally believed to be regulated only by vascular smooth muscle cells on arteries and arterioles supplying the brain, recent work has suggested a critical role for capillary pericytes, which are also contractile. This concept has evoked some controversy, especially over the relative contributions of arterioles and capillaries to the control of cerebral blood flow. Here we outline why pericytes are in a privileged position to control cerebral blood flow. First we discuss the evidence, and fundamental equations, which describe how the small starting diameter of

**Harvey Davis** is a Sir Henry Wellcome Fellow investigating the role of pericytes in diabetic neuropathy. **David Attwell** is the Jodrell Professor of Physiology at UCL. In recent years his lab has provided evidence for an important role of pericytes in regulating blood flow in the brain, heart and kidney.

capillaries, compared to upstream arterioles, confers a potentially greater control by capillary pericytes than by arterioles over total cerebral vascular resistance. Then we suggest that the faster time frame over which low branch order capillary pericytes dilate in response to local energy demands provides a niche role for pericytes to regulate blood flow compared to slower responding arterioles. Finally, we discuss the role of pericytes in capillary stalling, whereby pericyte contraction appears to facilitate a transient stall of circulating blood cells, exacerbating the effect of pericytes upon cerebral blood flow.

(Received 6 February 2023; accepted after revision 4 April 2023; first published online 9 April 2023)

**Corresponding authors** H. Davis and D. Attwell: Department of Neuroscience, Physiology & Pharmacology, University College London, London, UK. Email: harvey.davis@ucl.ac.uk, d.attwell@ucl.ac.uk

**Abstract figure legend** Graphical abstract indicating the three main principles by which pericytes provide a major contribution to the regulation of blood flow: (1) capillaries have smaller diameters than arterioles, (2) low branch order pericytes dilate faster than arterioles, and (3) pericyte-mediated capillary constriction can cause capillary occlusion by neutrophils or other blood cells. This figure was created using biorender.com.

## Introduction

When considering the regulation of tissue blood flow, there is increasing controversy regarding the relative contribution of larger muscular-walled vessels, such as arteries and arterioles, and smaller capillaries the tone of which is controlled by a network of pericytes. In small vessels such as capillaries, vessel dilatation or constriction can seem vanishingly small, at the limit of resolution of light microscopy, raising the question of how significantly such diameter changes could contribute to blood flow changes. In addition, the work of Burton (1951) showed that at low pressures within vessels (as might occur in capillaries), vessels can be unstable and collapse completely, and that a particular combination of elastic and active tension in a vessel wall is needed to allow graded control of diameter by contractile mural cells. Empirically, however, *in vivo* brain capillaries are patent, and they have their diameter controlled in a graded manner by pericytes, both for neuronal activity-evoked or pharmacologically-evoked dilatation (Hall et al., 2014; Korte et al., 2022) and for Alzheimer's disease-evoked or optogenetically-evoked constriction (Hartmann et al., 2021; Nortley et al., 2019). In this Review we use the latest literature to compare the theoretical and observed effects of small changes in capillary diameter and define how this informs our understanding of blood flow control. We focus mainly on the best studied pericytes – those in the cerebral microcirculation.

Regulation of local blood flow is largely achieved via changes in vessel lumen diameter, a major determinant of vascular resistance to blood flow. While it used to be thought that the lumen diameter was regulated by metabolic factors, such as a fall of $O_2$ or glucose level or a rise of $CO_2$ level, for the brain it has become clear that most ($\sim$70%) of the regulation reflects the neuronal activity-evoked release of factors such as nitric oxide and prostaglandin $E_2$ mediated by calcium concentration rises in astrocytes and neurons (reviewed by Attwell et al., 2010). In large arteries and arterioles, lumen diameter control is achieved by adjacent rings of smooth muscle cells, wrapped like bandages around the underlying endothelial cell-lined vessel (Fig. 1*A* and *B*). In capillaries, the smaller vessels which sprout from arterioles to deliver nutrients to the bulk of the surrounding tissue, diameter is instead controlled by pericytes, which bear functional similarities to smooth muscle cells, but are spatially separated and either exhibit a distinct 'bump-on-a-log morphology' (Fig. 1*B*) or are present at capillary junctions. Both smooth muscle cells and pericytes can control lumen diameter through constriction, or by relaxing and thus

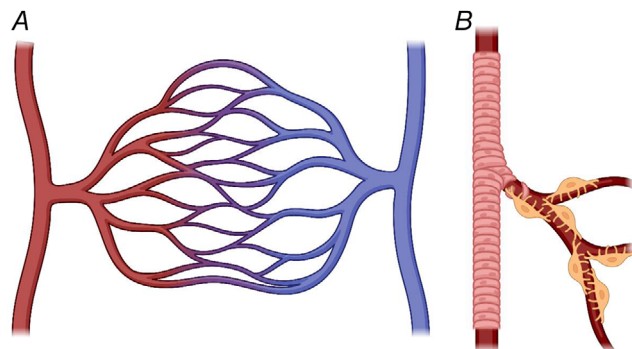

**Figure 1. Typical arrangement of the microvasculature**
*A*, a simple diagram showing the transit and direction of blood flow from arterioles (red, on the left) which progress into a larger capillary bed before ending in venular output (blue, on the right). *B*, cartoon illustrating the arrangement of smooth muscle cells around an arteriole (pink), in contrast to the pericyte-covered capillary bed with thin pericyte processes extending along and around the vessel (pericytes shown in yellow). Produced using BioRender.

dilating the underlying vessel's lumen. Accordingly, both cell types express contractile machinery, which is typically assessed experimentally via measurement of smooth muscle actin ($\alpha$-SMA) protein or transcript expression. Smooth muscle $\alpha$-SMA level is generally accepted as being higher in smooth muscle cells than in pericytes, a story which is further complicated by differences amongst the pericytes which line the capillary bed.

In central nervous system pericytes, perhaps the best described pericyte population, the quantity of $\alpha$-SMA expression negatively correlates with capillary branching order (i.e. expression is lower in capillaries that are more branches downstream from the originating arteriole; Alarcon Martinez et al., 2018; Hall et al., 2014; Hartmann et al., 2021; Rungta et al., 2018). This aligns with observations that lower order pericytes (branch order 1–4) can elicit larger and faster lumen diameter changes than higher branch order pericytes (branch order 5–9) (Hall et al., 2014; Hartmann et al., 2021). Although it is often assumed that this relationship reflects differences of pericyte $\alpha$-SMA protein expression alone, it is not understood whether there is a linear relationship between $\alpha$-SMA expression and contractility in pericytes. Indeed, prior work has detected an ability for higher order pericytes to reduce lumen diameter, despite the difficulty of detecting high levels of $\alpha$-SMA expression in these cells (Hartmann et al., 2021). Furthermore, the Lauritzen group has also named a population of capillary pericytes situated at the start of first-order capillaries as capillary 'sphincters', due to their unique morphology, spatially localised constriction and privileged position at the junction between arterioles and the capillary bed (Grubb et al., 2020). However, these cells are functionally similar to other pericytes and, as contractile cells residing within the capillary bed, sphincters fit within the original definition of pericytes (Zimmerman, 1923). Further, despite their potential physiological importance, only 28% of first order capillaries were demonstrated to exhibit sphincter-like pericytes (Grubb et al., 2020), which are therefore a minor population. For this reason, the term pericyte is used here to refer to all contractile cells residing in the capillary bed.

In order for these perivascular cells to control local blood flow, smooth muscle cells and pericytes are regulated by various direct and indirect factors, including nitric oxide, prostaglandin $E_2$, 20-hydroxyeicosatetraenoic acid and noradrenaline (Hall et al., 2014; Korte et al., 2023; Mishra et al., 2016). Together, these confer a dynamic and often rapid control of blood vessel diameter, and consequently vascular resistance and local blood flow, in response to changes in physiological parameters such as neural activity. In a crude but clear demonstration of the power of these mechanisms, topical application or local injection of endothelin-1 (a potent vasoconstrictor) into the cerebral cortex causes a localised reduction of blood flow sufficient to cause ischaemic tissue damage downstream of reduced local blood flow (Windle et al., 2006). This dramatic control is also observed in pathology. For example, in ischaemic stroke pericyte constriction (and death) is sufficient to prevent or limit local tissue reperfusion following restoration of larger vessel blood flow, in a manner reversible by pharmacologically or genetically preventing arteriolar and capillary constriction (Korte et al., 2022).

However, there exists disagreement as to the relative importance of changes in vascular diameter by smooth muscle cells *versus* pericytes. Principally, as the absolute lumen diameter changes observed in arterioles are generally larger, owing to the larger dimensions of the arteriole itself (Hartmann et al., 2021), many investigators have been led to disregard the effects produced by small capillary constrictions or dilatations. This mistaken interpretation, compounded by disagreements over the identity and contractile ability of pericytes, as discussed previously (Attwell et al., 2016), has led some to disregard recent literature focusing on pericyte-mediated changes in blood flow. We believe that this is a barrier to advancing our collective understanding of the vasculature, and so we have attempted here to briefly summarise the arguments supporting the importance of pericyte-mediated diameter changes. For clarity, we can break our argument down into three major points:

(i) Small diameter changes have large effects. The relatively small lumen diameter changes elicited by pericytes can be theoretically and experimentally demonstrated to have a disproportionate effect upon overall vascular resistance and blood flow (as a result of both Poiseuille's law and the dependence of blood viscosity on diameter), and are thereby critical to vascular function.

(ii) All about timing. In low branch order capillaries, pericyte-mediated lumen diameter changes are quicker than in arterioles, thereby providing an additional biological function.

(iii) Neutrophils lend a hand. Even deceptively small absolute capillary diameter changes can cause capillary occlusion by neutrophils and red blood cells, halting blood flow.

**Small diameter changes have large effects.** Pericytes of all branch orders appear to be able to regulate capillary lumen diameter (Hall et al., 2014; Hartmann et al., 2021; Mughal et al., 2023), but the absolute changes are often small. As experimental evidence seems to suggest that these small changes are profoundly important, we attempt here to explain these changes using a simple mathematical framework.

In its simplest form, the relationship between blood flow and diameter can be described by Poiseuille's law, where the flow through a vessel ($Q$) is given by the pressure difference across the vessel ($\Delta P$) divided by the vascular resistance ($R$).

$$Q = \Delta P/R \qquad (1)$$

For laminar flow of a pure liquid, the resistance ($R$) is defined in terms of the vessel diameter ($d$), vessel length ($L$) (defined here as the length of vessel controlled by each pericyte and/or smooth muscle cell) and viscosity ($\eta$) as:

$$R = \frac{128 \times \eta \times L}{\pi \times d^4} \qquad (2)$$

While blood viscosity ($\eta$) is often regarded as constant, experimental observations suggest that viscosity increases greatly in small diameter vessels, including the lumen diameter range observed in capillaries. As reviewed previously, this is due to the presence of deformable blood borne cells (Secomb & Pries, 2013). The relationship between vessel diameter ($d$) and viscosity ($\eta$) has previously been described (Hirunpattarasilp et al., 2022) in the diameter range between 3 and 8 $\mu$m as being given by $\frac{A}{d^n}$, where $A$ and $n$ are constants, with $n = 1.647$ giving a reasonable fit to experimental data (for a blood haematocrit of 0.45).

A final complication arises from the nature of pericyte contraction, where vessel lumen diameter changes are greater under the pericyte soma than under the distal pericyte processes, because there are more circumferential processes near the soma (Nortley et al., 2019). Experimental data from the brain and the kidney suggest that, in the presence of a vasoconstrictor, the capillary lumen diameter has a near linear relationship with distance from the pericyte soma (Nortley et al., 2019, Figs 4*d*, 5*f* and S3*b*; Freitas & Attwell, 2022, Fig. 5*d*), as shown in Fig. 2. For simplicity we will initially assume that, when pericytes are constricting capillaries, the capillary diameter reaches its maximum value exactly midway between adjacent pericytes (this assumption will be relaxed later). The lumen diameter at the pericyte soma, where the pericyte's constricting influence is greatest, is referred to as $d_1$ (Fig. 2). The lumen diameter at the point midway between the somata of adjacent pericytes is referred to as $d_2$ and the distance of this point away from the pericyte soma as $L$. The dependence of lumen diameter ($d$) on distance from the pericyte soma ($x$) can thus be described with the following linear relationship:

$$d(x) = d_1 + \left(\frac{d_2 - d_1}{L}\right) \times x \qquad (3)$$

and the total resistance can be calculated as:

$$\int_0^L \frac{A}{d^n} \times \frac{128}{\pi d^4} \, dx \qquad (4)$$

When integrated, and then normalised by the resistance obtained when the capillary diameter is $d_2$ everywhere, this produces the following equation (Hirunpattarasilp et al., 2022) for the ratio of the resistance when the pericyte is constricted to that when the diameter is uniform:

$$R_{constricted}/R_{uniform} = \left[1 - (d_2/d_1)^{(3+n)}\right] / \\ \left[((d_1/d_2) - 1) . (3 + n)\right] \qquad (5)$$

In fact, in the absence of vasoconstrictors, the capillary diameter is not uniform but is larger at the pericyte soma than at a point midway between pericytes, possibly because the presence of the pericyte induces growth of the endothelial tube (Hall et al., 2014). For this situation, eqn (5) can be used to calculate the ratio of the resistance in this relaxed state relative to the value when the diameter is $d_2$ everywhere. Then, dividing the ratio obtained from eqn (5) when the pericytes are constricting to the ratio obtained when they are relaxed gives the factor by which the capillary resistance is increased during constriction relative to the relaxed state. We term this factor $F_{cap}$.

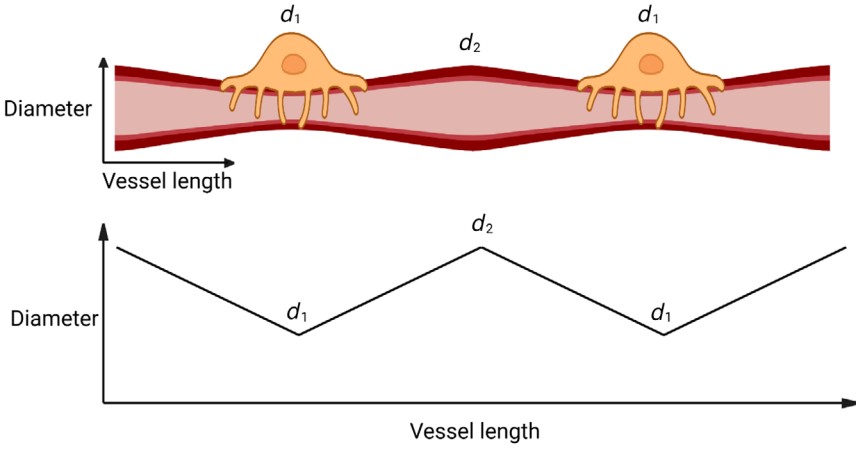

**Figure 2. Assumed relationship between vessel diameter and position along a capillary in a pericyte-controlled capillary bed**
Based upon experimental evidence, pericyte constriction of the underlying capillary is assumed to be greatest at the pericyte soma and to decrease linearly with distance over half the interpericyte distance. Produced using BioRender.

**Table 1. The effect of pericyte mediated capillary constrictions observed in pathology upon vascular resistance and flow**

| Pathology | $d_1/d_2$ (baseline) | $d_1/d_2$ (pathology) | Capillary bed resistance ratio | Total vascular resistance ratio | Flow ratio | Reference |
|---|---|---|---|---|---|---|
| Covid-19* (brain slice) | 1.078 | 0.963 | 1.371 | 1.212 | 0.825 | Hirunpattarasilp et al. (2023) |
| Stroke (CCAO) (*in vivo*) | 1.200 | 1.020 | 1.308 | 1.308 | 0.765 | Korte et al. (2022) |
| Hyperoxia (brain slice) | 1.078 | 0.770 | 1.988 | 1.988 | 0.503[†] | Hirunpattarasilp et al. (2022) |
| Alzheimer's (*in vivo*) | 1.270 | 0.700 | 3.684 | 3.684 | 0.271[†] | Nortley et al. (2019) |

The table shows, for different pathologies (left column), the values of $d_1/d_2$ in control conditions and in the pathology (where $d_1$ is the capillary diameter at the soma, and $d_2$ is the diameter at the midpoint between pericytes), the resulting factor increase in the resistance of the capillary bed (from eqn (5)), the factor by which the total vascular bed resistance (arterioles + capillaries + venules) increases (from eqn (6)), and the resulting effect on flow through the whole vascular bed. *Covid-19 was modelled by the presence of the spike protein receptor binding domain. Covid-19 differences were taken as angiotensin with pseudovirus vs. angiotensin without pseudovirus. †Prediction is a greater reduction than reported in the original paper because here we additionally included the effect of the viscosity change.

This calculated factor can be used to demonstrate the importance of small constrictions mediated by capillary pericytes, by calculating the change in total brain vascular resistance following an observed capillary constriction. The total intra-parenchymal resistance (to which blood flow will be inversely proportional if a constant pressure difference is applied by the pial vessels at the top of the penetrating arteriole and ascending venule) is given by:

$$R_{\text{total}} = R_{\text{arteriole}} + R_{\text{capillaries}} + R_{\text{venule}}$$

where the values of $R_{\text{arteriole}}$, $R_{\text{capillaries}}$ and $R_{\text{venule}}$ are assumed to be in the ratio 1:4:2 when pericytes are relaxed (for blood flow reaching cortical layer 4: Blinder et al., 2013). We consider only the vessels within the brain parenchyma that can be signalled to by the neurons and associated cells in order to change their energy supply, that is the penetrating arterioles, capillaries and ascending venules. Blinder et al. (2013) give the resistance of the penetrating arteriole and ascending venule between the pia and layer 4 as 0.1 and 0.2 $\text{P}/\mu\text{m}^3$, while the resistance between capillaries joining arterioles and venules in layer four was taken as the asymptotic network resistance in their Fig. 2*g*, that is 0.4 $\text{P}/\mu\text{m}^3$ (see also Gould et al., 2016). This 1:4:2 ratio may differ significantly for other vascular beds. Using this methodology, we can then investigate the importance of changes of capillary resistance for changing the total blood flow, by calculating the factor by which pericyte contraction alters the total parenchymal resistance, as follows:

$$F_{\text{Total}} = \frac{R_{\text{arteriole}} + F_{\text{cap}} \times R_{\text{capillaries}} + R_{\text{venule}}}{R_{\text{arteriole}} + R_{\text{capillaries}} + R_{\text{venule}}} \quad (6)$$

Importantly, as considered in this section, there are conditions for which only the capillary resistance is altered; for example, in Alzheimer's disease model mice, pericytes are constricted while arterioles and venules do not change their diameter (Nortley et al., 2019, Fig. 5). Nevertheless, it will often be the case that arterioles and capillaries are dilated or constricted at the same time: we deal with this situation below.

In Table 1 we have given literature values for the ratio of $d_1/d_2$ in the relaxed state and in various pathologies (from the papers cited). From these we calculate the factor by which the resistance of the capillary bed is increased in pathology, and hence (assuming no change of the resistance of the arteriole and venule segments of the vasculature) the factor ($<1$) by which the flow through the whole vascular network is multiplied for each pathology.

Thus, with pericyte-mediated capillary constrictions of the magnitude seen experimentally either in brain slices or *in vivo*, cerebral cortical blood flow is predicted to be reduced between 17% (for Covid-19) and 73% (for Alzheimer's disease). These reductions are potentiated significantly by the increase of viscosity induced by the reduction of vessel diameter, without which the reductions would range between 13% and 57% (and, as described below, they are potentiated further by cells occluding the constricted vessels).

The analysis above assumes that there are no spaces along the capillary between pericytes where there is no constriction. This is a good assumption for pericytes on the first few branch orders of the capillary bed (where much of the total capillary resistance is located because for higher branch orders there are many more capillary

**Table 2. Mean cortical inter-pericyte distances observed in rat, mouse and human tissue**

| Species | Rat | Mouse | Human |
|---|---|---|---|
| Interpericyte distance ($\mu$m) | 45.5 | 20–90 | 65.3 |
| Paper | Hall et al. (2014) | Grant et al. (2019), Shaw et al. (2022) | Nortley et al. (2019) |

**Table 3. Flow ratio across whole vascular bed when pericytes are <40 $\mu$m apart or are 90 $\mu$m apart**

| Pathology | Flow ratio with pericyte separation <40 $\mu$m | Flow ratio with pericyte separation 90 $\mu$m |
|---|---|---|
| Covid-19 (brain slice) | 0.825 | 0.914 |
| Stroke (CCAO) (*in vivo*) | 0.765 | 0.880 |
| Hyperoxia (brain slice) | 0.503 | 0.695 |
| Alzheimer's (*in vivo*) | 0.271 | 0.456 |

segments in parallel; see below), where it has been shown that the separation of pericyte somata is of the order of 20−40 $\mu$m (Grant et al., 2019; Shaw et al., 2022). However, on higher branch order capillaries, pericyte somata are further apart, perhaps ∼90 $\mu$m (Table 2; Grant et al., 2019) and so, if capillary constriction occurs for a distance up to 20 $\mu$m on either side of the soma (Nortley et al., 2019), there could be a 50 $\mu$m length of capillary in between the ends of the pericyte processes where no constriction occurs at all (Fig. 3). This situation can be modelled by replacing eqn (6) with:

$$F_{\text{Total}} = \frac{R_{\text{arteriole}} + \left(\frac{40.F_{\text{cap}}+50}{90}\right) \times R_{\text{capillaries}} + R_{\text{venule}}}{R_{\text{arteriole}} + R_{\text{capillaries}} + R_{\text{venule}}}$$

As shown in Table 3, even when the influence of pericytes is limited to a shorter length (20 $\mu$m each side of the soma), where pericytes are experimentally observed to exert their greatest effect (Nortley et al., 2019), they still have a large potential influence over the total vascular resistance observed in the cortex. Predicted flow changes when pericytes are constricted by the various disorders in Table 1 range from an 8.6% reduction in Covid-19 to a 54% reduction in Alzheimer's disease.

We have established that small changes in capillary diameter have large effects upon calculated vascular resistance. However, these simple formulations rely upon several assumptions, principally that (i) pericyte density is similar across the vascular bed, (ii) the magnitude of capillary diameter changes is uniform across the capillary bed, and (iii) there is no associated change in arteriolar diameter. Our simple model would be further complicated by these considerations. Pericyte density and morphology change across the capillary bed, with less circumferential processes and less $\alpha$SMA expression in higher branch order pericytes, thus favouring a reduced control of capillary diameter in those branches. However, a simple geometrical argument suggests that adjusting the diameter of the first few capillary branch orders from a penetrating arteriole should have a much larger effect on flow than would adjusting the diameter of high

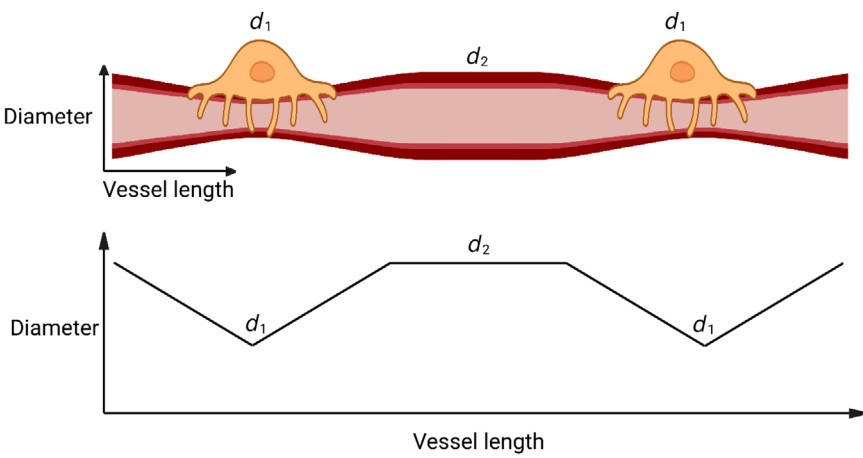

**Figure 3. More realistic relationship between vessel diameter and position along vessel in a pericyte-controlled capillary bed**

In this case, pericytes are assumed to impose a constriction which is greater at the soma midline, and decreases to a point 20 $\mu$m away, beyond which circumferential processes and observable vessel constriction are usually absent. The vessel length in between the ends of the pericyte processes is assumed to have a constant diameter $d_2$. Produced using BioRender.

branch order capillaries. This can be seen as follows, by assuming simplistically that each capillary segment has the same length and diameter. At each capillary branch the area available for blood flow thus doubles and the resistance for the following pair of segments therefore halves. Consequently, the sum of the resistances of the first, second, third, etc. branch order capillaries in this idealised network is proportional to the sum of:

$$1 + \frac{1}{2} + \frac{1}{4} + \frac{1}{8} + \frac{1}{16} \cdots$$

In the first four branch orders this factor sums to $1^7/_8$, which is within 6% of the value (i.e. 2) for an infinitely branched network of this sort (the higher order branches contribute less resistance because there are so many of them in parallel), implying that pericytes on the first few branch orders (or indeed the final few branches adjoining the ascending venule) should have a disproportionate role in adjusting the total resistance and hence flow. Tempering this argument is the fact that capillary diameters decrease at higher branch orders (Hartmann et al., 2021), which will increase the resistance of each segment, both directly because of the reduced area for flow and indirectly because of the increase in blood viscosity at smaller diameters (see above). Nevertheless, this weighting towards the resistance of the first few capillary branches, which is where pericyte properties have been best studied, suggests that our modelling may capture reasonably well the importance of pericytes in controlling cerebral blood flow.

This model has not considered that pericytes (and arteriolar smooth muscle) do not consistently respond to stimuli (Hall et al., 2014; Nortley et al., 2019; Shaw et al., 2022). This may influence the role of pericytes across the capillary bed, but it is worth noting that no significant difference in response rate has been reported between capillary branch orders 1−4 (Shaw et al., 2022) and so this relationship may be uniform across the capillary bed.

It is also important to compare our predictions for the influence of pericytes, with the effect of resistance changes caused by arteriolar constriction, or with both arterioles and capillaries constricting together. As arterioles are typically larger than the 3−8 μm range for which we use (above) an approximate equation to describe capillary viscosity, we used the more general eqns (6)−(7) from Secomb and Pries (2013) to calculate arteriole viscosity. We then calculated the ratio ($F_{art}$) between the constricted arteriolar resistance and the baseline arteriolar resistance to derive the factor change in the total vascular bed resistance as:

$$F_{Total} = \frac{F_{art} \times R_{arteriole} + R_{capillaries} + R_{venule}}{R_{arteriole} + R_{capillaries} + R_{venule}}$$

or, for a calculation with both arterioles and capillaries constricting (with $F_{cap}$ defined above):

$$F_{Total} = \frac{F_{art} \times R_{arteriole} + F_{cap} \times R_{capillaries} + R_{venule}}{R_{arteriole} + R_{capillaries} + R_{venule}}$$

The resulting factors for change of blood flow will be the inverse of the values given for the factors by which the total resistance increases. The results in Table 4 show that, for a given percentage constriction, at the pericyte somata or along whole arterioles, the effect of capillary constriction is greater than that of arteriole constriction.

**All about timing.** The time course over which pericytes change capillary lumen diameter gives pericytes a key role in regulating rapid blood flow changes. First and second order capillary dilatations (including those mediated by contractile cells on first order capillaries sometimes referred to as sphincters; Grubb et al., 2020) are significantly faster than in arterioles (Hall et al., 2014; Khennouf et al., 2018), implying that pericytes can respond first to physiological energy demands such as neuronal activity. This implies that first and second order pericytes have a somewhat privileged role in controlling local energy supply through dilatation, which may confer a unique and important physiological role. In contrast, higher branch order pericytes may change the capillary diameter with a slower time course (Hartmann et al., 2021), although they will still contribute to an overall change in resistance as explained above.

**Neutrophils lend a hand.** In addition to the substantial increase in vascular resistance and decrease in blood flow predicted from the decrease in vessel diameter and associated rise in blood viscosity that follows pericyte constriction, pericyte constriction can also cause capillary blockage. This is particularly profound for the pericytes around the descending vasa recta of the kidney, where ischaemia-evoked blockage occurs in roughly 80% of capillaries following a diameter reduction of only ∼30%, which should not by itself be sufficient to prevent blood flow (Freitas et al., 2022). Similar blockages occur after cerebral ischaemia, and in diabetes and Alzheimer's disease (Cruz Hernández et al., 2019; Korte et al., 2022; Sharma & Brown, 2022) although they are less severe. They are associated with neutrophils (and also monocytes and red blood cells: Korte, N., Barkaway, A. & Attwell, D., unpublished) becoming lodged near contracted pericytes (El Amki et al., 2020; Cruz Hernández et al., 2019; Korte et al., 2022), as a result of the relatively less deformable neutrophils being unable to pass through the constricted part of the capillary. Thus, pericyte constriction facilitates neutrophil-mediated capillary blockage.

**Table 4. A comparison of the effect of arteriole or capillary constriction, alone or combined, upon total vascular resistance**

| % change in diameter (whole arteriole or at pericyte soma) | Total resistance change ratio (% flow decrease) | | |
|---|---|---|---|
| | Arteriole constriction | Capillary constriction (as in Table 1) | Combined constriction |
| 10% | 1.097 (8.8%) | 1.185 (16%) | 1.282 (22%) |
| 20% | 1.286 (22%) | 1.488 (33%) | 1.773 (44%) |
| 30% | 1.688 (41%) | 2.024 (51%) | 2.713 (63%) |
| 40% | 2.660 (62%) | 3.083 (68%) | 4.744 (79%) |

Effect on the total resistance of the vascular bed (arteriole + capillaries + venule), and on blood flow through the vascular bed (shown in brackets as a percentage decrease), of constriction of either the whole arteriole, or of capillaries (separated by 40 $\mu$m) at pericyte somata, or of both, by the percentages shown in the left column. For these calculations, arterioles were assumed to have a baseline diameter of 12.4 $\mu$m (Hall et al., 2014), and the baseline dilatation at pericyte somata was set to the average of the values in Table 1 (1.157). All calculations used Poiseuille's law and the dependence of viscosity on vessel diameter.

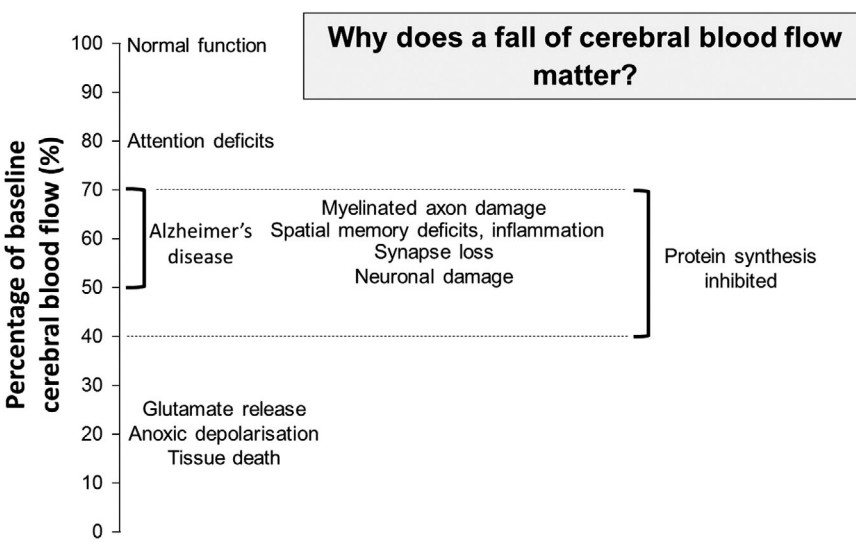

**Figure 4. Consequences of different percentage falls of cerebral blood flow**
Schematic representation based on patient studies of ageing humans; animal experiments with coils placed on arteries to the brain giving a 30% cerebral blood flow (CBF) fall; 2-vessel occlusion experiments in rodents giving a ~50% CBF fall; middle cerebral artery occlusion in rodents producing at least an ~80% fall in CBF.

## Effect of capillary-mediated cerebral blood flow decreases on brain function

To assess the consequences of the capillary-mediated decreases in cerebral blood flow, in Fig. 4 we have pulled together data from numerous papers to give a summary of the functional effects of different percentage reductions in flow. It can be seen that the pericyte-mediated reductions of flow predicted in Table 1 (17−73%) are predicted to seriously impact on brain function.

## Summary and possible therapeutic implications

We have outlined how small changes in capillary diameter associated with pericyte constriction are not only physiologically relevant, but may in pathological conditions dominate the control of tissue blood flow by virtue of the reduction in vessel lumen size, an associated increase in blood viscosity at small diameters, and a resulting promotion of capillary block by circulating cells, especially neutrophils. This new understanding of how blood flow is restricted in pathology may stimulate new therapeutic approaches based on preventing pericyte constriction or capillary block, as has already been shown to have positive outcomes in animal models of cerebral, renal and cardiac ischaemia, and Alzheimer's disease (El Amki et al., 2020; Freitas et al., 2022; Cruz Hernández et al., 2019; Korte et al., 2022; O'Farrell et al., 2017).

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

## Additional information

### Competing interests

None.

### Author contributions

H.D. 60%, D.A. 40%. Both authors have read and approved the final version of this manuscript and agree to be accountable for all aspects of the work in ensuring that questions related to the accuracy or integrity of any part of the work are appropriately investigated and resolved. All persons designated as authors qualify for authorship, and all those who qualify for authorship are listed. For the purposes of open access, the authors have applied a CC-BY public licence to any Author Accepted Manuscript version arising from this submission.

### Funding

A Wellcome Trust Sir Henry Wellcome fellowship (224049) to H.D.; ERC (740427) and Wellcome Trust (219366) Investigator Awards and a Rosetrees Trust grant (M153-F2) to D.A.

### Keywords

blood flow, capillary, pericyte, resistance, viscosity

### Supporting information

Additional supporting information can be found online in the Supporting Information section at the end of the HTML view of the article. Supporting information files available:

**Peer Review History**

