## [Peer Review History · The Journal of Physiology]

A tight squeeze: how do we make sense of small changes in microvascular diameter?

Harvey Davis and David Attwell
DOI: 10.1113/JP284207

Corresponding author(s): Harvey Davis (harvey.davis@ucl.ac.uk)

Review Timeline:

Submission Date:	06-Feb-2023
Editorial Decision:	08-Mar-2023
Revision Received:	02-Apr-2023
Accepted:	04-Apr-2023

Senior Editor: Peter Kohl

Reviewing Editor: Peter Kohl

Transaction Report:

Dear Dr Davis,

Re: JP-OP-2023-284207 "A tight squeeze: how do we make sense of small changes in microvascular diameter?" by Harvey Davis and David Attwell

Thank you for submitting your manuscript to The Journal of Physiology. It has been assessed by a Reviewing Editor and by 2 expert referees and we are pleased to tell you that it is acceptable for publication following satisfactory revision.

Please advise your co-author of this decision as soon as possible.

REVISION CHECKLIST:

We look forward to receiving your revised submission.

Yours sincerely,

Peter Kohl
Senior Editor
The Journal of Physiology

EDITOR COMMENTS

The critical review by external peers illustrates that the topic is relevant and the paper is controversial - two of the requirements for an Opinion type article in JP. We would like to encourage the authors to consider the feedback received, to improve the manuscript. In particular, we would like to suggest that the controversies in the subject area (pre-capillary sphincters versus pericytes) are addressed even more explicitly and head-on.

REFEREE COMMENTS

Referee #1:

This Opinion article is from a leading laboratory known internationally in the field of cerebral flow regulation. This manuscript reviews key findings that support a role for capillary regulation of blood flow to the brain via pericyte constriction or dilation. This is a controversial and timely topic with a growing cadre of active investigators. Extensive explanations with equations linking diameter and flow are shown to help explain how general and local factors influence flow and resistance. While this Opinion piece nicely addresses the role of pericytes in regulating cerebral flow, there are some concerns that dampen enthusiasm.

Major issues:

- 1. This review comes on the heels of a review of pericytes in cerebral flow regulation earlier this year published in J. Phys (Mughal, Nelson, Hill -Eubanks. J.Physiol, February 2023). This published review covers many of the same items covered by the current Opinion piece.
- 2. The authors use phraseology indicating that capillary flow regulation contributes more than arteriolar regulation of cerebral blood flow. This is highly unlikely for several reasons. First measurements of pressure within different segments of the circulation in vivo show that the largest drop in pressure occurs across the arteriolar bed (Chilian et al. Circulation Research vol-66 p-1227 1990). Having this baseline tone is what allows the vessel to either dilate or constrict depending on the need. In most cases it is metabolism that promotes the need for vasodilation and arterioles are programmed to respond with relaxation and initiation of flow-induced dilation to recruit upstream vessels in the delivery of oxygen and nutrients. Third, pressure inside capillaries is maintained at a fairly low level so there's little to no opportunity for increasing flow by relaxing pericyte constriction. Data seem more convincing that arterioles, and some larger arteries in the brain regulate neurovascular coupling and cerebrovascular perfusion. It is more likely that delivery of blood to the brain is mediated by arterioles but local capillary distribution can be modified by pericyte constriction. Or as the authors suggest, the rapidity of the pericyte response might usher in a vasoconstriction or dilation that is taken over once the smooth muscle cells begin to contract or relax.

Other

- How was it determined in fig 6 that the ratio of resistance between arterioles capillaries and venules is 1:4:2. Based on measured pressure drop along the arterial-capillary-venular tree I would expect the ratio to be more like 8:2:1.
- On pages 4-6 the argument is made that calculating F_{cap} (the ratio of a ratio) can determine the extent to which pericytes are playing a role in cerebral flow. It is not clear how the proposed means of changing pericyte activity would not also affect arterioles either directly or indirectly.
- On page 9 it is stated that in prior studies, responses in capillaries in 1-4 branch order are similar. How can that be if at the same time the density of pericytes decreases dramatically in larger vs. smaller branch order capillaries?
- It is stated that very small changes in diameter have dramatic effects on flow. This is correct, is explained in the Poiseuille equation, and applies to all vessels not just small ones. Post-arteriolar and arteriolar blood flow distribution should include a description of Poiseuille's equation relating resistance as proportional to the change in radius r^4 . Therefore a 50% constriction yields a 94% reduction in flow. In relation to Fig.4 the data shown reveal a much more attenuated response likely due to autoregulation or other compensatory mechanisms. Page 10: What is the teleological explanation for ischemic activation of vessel obstruction by circulating neutrophils?

Referee #2:

The paper entitled: A tight squeeze: how do we make sense of small changes in microvascular diameter? summarizes a point of view regarding the role for pericytes on brain capillaries for control of the cerebral circulation. The pericyte topic and the importance of distinguishing between arterioles and capillaries was recently reviewed in an excellent paper in The Journal of Physiology by Mughal et al. and this paper follows in the footsteps of that paper.

For about a decade it has been documented that pericytes contribute to an important part of cerebrovascular resistance and this paper adds to the existing evidence using a modeling approach. The model is mainly based on excellent results obtained in the Attwell lab, mainly in slices but also in vivo and the model faithfully reproduces the observations that the Attwell group have reported in several publications. The model is simple and is based on the notion that pericytes constrict the underlying capillary close to a pericyte soma and that the diameter increases linearly between cell bodies. It is unclear whether this assumption is canonical in the pericyte/capillary field and to what extent this has been documented. This may

be important because that notion is the basis for the extended version of the elegant model that is proposed by the authors.

A second issue is that the description of the vascular network and the model do not include variations of the influence of pericytes across the microvascular bed, e.g., the very dynamic precapillary sphincters at the inlet and the very slow reacting pericytes at the venous end. The text also does not relate to previously published models of regulation of brain capillaries based on simple hemodynamic network models nor complex models of transit times. Still, this text is a valuable contribution to the pericyte literature giving novel insight to cerebrovascular resistance exerted by the capillary network in health and even more so in disease states.

END OF COMMENTS

Confidential Review

06-Feb-2023

Dear Peter,

JP-OP-2023-284207 "A tight squeeze: how do we make sense of small changes in microvascular diameter?" by David Attwell and Harvey Davis

We are now submitting a new version of this Opinion piece, which has been revised as discussed below in response to the comments of the referees.

Best wishes,

Harvey Davis & David Attwell

Editor's comments

The critical review by external peers illustrates that the topic is relevant and the paper is controversial - two of the requirements for an Opinion type article in JP. We would like to encourage the authors to consider the feedback received, to improve the manuscript.

We thank the Editor for these encouraging comments.

In particular, we would like to suggest that the controversies in the subject area (pre-capillary sphincters versus pericytes) are addressed even more explicitly and head-on.

The authoritative original definition of capillary pericytes (by Zimmerman in 1923) included all contractile cells found on capillaries, between the arterioles and the venules. This includes those evoking a cuff-like constriction of the 1st order capillaries (branching off arterioles) that were characterised in the brain by Martin Lauritzen's group (Grubb et al., 2020) and termed "sphincters". Grubb et al. quantified that only 28% of 1st order capillaries had a "sphincter", so they are a minority population of pericytes even on the 1st order capillaries that they are found on, but their position on the 1st order capillary gives them more weight for affecting blood flow than if they were on higher order capillary branches. Although the constriction evoked by "sphincters" may be more spatially localised than that evoked by other pericytes, there is no fundamental difference in their mode of operation to other pericytes on the 1st-4th branch orders. We have now explained all of this on pages 3-4 of the paper.

REFEREE COMMENTS

Referee #1:

This Opinion article is from a leading laboratory known internationally in the field of cerebral flow regulation. This manuscript reviews key findings that support a role for capillary regulation of blood flow to the brain via pericyte constriction or dilation. This is a controversial and timely topic with a growing cadre of active investigators. Extensive explanations with equations linking diameter and flow are shown to help explain how general and local factors influence flow and resistance.

We thank the referee for their positive comments.

While this Opinion piece nicely addresses the role of pericytes in regulating cerebral flow, there are some concerns that dampen enthusiasm.

Major issues:

- 1. This review comes on the heels of a review of pericytes in cerebral flow regulation earlier this year published in J. Phys (Mughal, Nelson, Hill -Eubanks. J.Physiol, February 2023). This published review covers many of the same items covered by the current Opinion piece.

In fact the Mughal et al. review was published the day after we submitted our invited Opinion piece, and so the content of our piece was conceived independently of that work. We agree that Mughal et al. (2023) is a timely review on cerebral blood flow regulation, and we have now added a reference to it on page 5 of our paper. However, while that review focusses on the post-arteriole transition zone (the first few capillary branches off of a penetrating arteriole in the brain), the topic of our Opinion piece is completely different, i.e. how to estimate the effect on vascular resistance of small changes in capillary diameter generated by pericytes. Thus, our commentary is complementary to other reviews such as that of Mughal et al. (2023).

- 2.The authors use phraseology indicating that capillary flow regulation contributes more than arteriolar regulation of cerebral blood flow. This is highly unlikely for several reasons.

First measurements of pressure within different segments of the circulation in vivo show that the largest drop in pressure occurs across the arteriolar bed (Chilian et al. Circulation Research vol-66 p-1227 1990).

We have now added a more explicit statement that we focus mainly on the best studied pericytes - those in the cerebral microcirculation (on page 2, end of para 1). Furthermore, we are focussing on vessels which may receive a signal (from the cells of the tissue) that instructs them to adjust the blood flow to alter the energy supply, as we have now stated on page 7 (para 3). In the brain parenchyma that means the penetrating arterioles, capillaries and ascending venules, all of which have a diameter below ~15 microns (Hall et al., 2014, Table 1; Nortley et al, 2019, Fig 5h; both papers cited in this manuscript). These vessels are much too small to be studied using the micropuncture techniques that are used in the Chilian et al. paper cited by the referee, which is on large (~150 micron) arterioles and venules in the cardiac circulation. Thus, as demonstrated by the papers we cite on page 7, para 3 (Blinder et al., 2013, to which we have now added Gould et al., 2016), of the intra-parenchymal vessels in the brain, the capillaries contribute the most resistance.

Having this baseline tone is what allows the vessel to either dilate or constrict depending on the need.

We make no argument regarding the presence or absence of basal tone although this will indeed be present at some level in vivo in both pericytes and arterioles, as it is evoked by release of noradrenaline from axons of the the brain's locus coeruleus neurons (there is no sympathetic innervation of the brain parenchyma).

In most cases it is metabolism that promotes the need for vasodilation and arterioles are programmed to respond with relaxation and initiation of flow-induced dilation to recruit upstream vessels in the delivery of oxygen and nutrients.

In the cerebral circulation, although it was for some time believed (as suggested by the referee) that most neuronal activity-evoked vasodilation was triggered by metabolic factors (such as a fall of pO₂ or glucose level, or a rise of pCO₂), in the last 20 years it has become clear that most (~70%) of the vasodilation reflects the neuronal activity-evoked release of factors such as nitric oxide and prostaglandin E₂ mediated by calcium concentration rises in astrocytes and neurons (reviewed by Attwell et al., 2010, Nature 468, 232). We have now stated this on page 2 (last para).

Third, pressure inside capillaries is maintained at a fairly low level so there's little to no opportunity for increasing flow by relaxing pericyte constriction.

This comment may relate to the work of Burton (1951, Am J Physiol 164, 319) who showed: (i) that below a critical pressure within vessels it can occur that the vessels are unstable and are predicted to close completely, and (ii) that a particular combination of elastic tension and active tension in a vessel wall is needed to allow graded control of diameter by contractile mural cells. Empirically, however, in vivo brain capillaries are patent and we have shown that they do have their diameter controlled in a graded manner by pericytes, both for neuronal activity-evoked dilation (Hall et al., 2014, Nature [cited in our paper], see Fig. 3) and for pharmacologically-evoked dilation (Korte et al., 2022, JCI, see Fig. 7F) as well as for Alzheimer's disease-evoked constriction (Nortley et al., 2019, Science [cited in our paper], see Fig. 5). Similarly, elegant work by Andy Shih (Hartmann et al., 2021, Nature Neurosci, cited in our paper) using optogenetics to constrict pericytes has demonstrated that pericyte-mediated capillary constriction in vivo has a significant influence on blood flow (measured as red blood cell velocity), implying that pericyte relaxation will have the opposite effect. Thus, any dilation of capillaries will decrease the vascular resistance and thus increase cerebral blood flow (to an extent determined by the fraction of the total resistance that is in the capillaries, as opposed to in the arterioles and venules).

We have now added a section on this point to the first paragraph of the paper which states:

"In addition, the work of Burton (1951) showed that at low pressures within vessels (as might occur in capillaries) vessels can be unstable and collapse completely, and that a particular combination of elastic and active tension in a vessel wall is needed to allow graded control of diameter by contractile mural cells. Empirically, however, in vivo brain capillaries are patent, and they have their diameter controlled in a graded manner by pericytes, both for neuronal activity-evoked or pharmacologically-evoked dilation (Hall et al., 2014; Korte et al., 2022), as well as for Alzheimer's disease-evoked or optogenetically-evoked constriction (Nortley et al., 2019; Hartmann et al., 2021)."

Data seem more convincing that arterioles, and some larger arteries in the brain regulate neurovascular coupling and cerebrovascular perfusion. It is more likely that delivery of blood to the brain is mediated by arterioles but local capillary distribution can be modified by pericyte constriction. Or as the authors suggest, the rapidity of the pericyte response might usher in a vasoconstriction or dilation that is taken over once the smooth muscle cells begin to contract or relax.

The Referee seems to be basing their conclusion on the concept that most of the adjustable resistance in the tissue vascular bed is located in arterioles rather than capillaries. As detailed in the next point, this may be true for the heart, but is incorrect for the brain (see Blinder et al., 2013, cited in the paper, and also Gould et al. (2016, JCBFM 37, 52: "The capillary bed offers the largest hemodynamic resistance to the cortical blood supply" who conclude on page 65 that "the steepest pressure drops occur in the capillary bed") which we have now also cited on page 7. We also added a more explicit statement that we focus mainly on the best studied pericytes - those in the cerebral microcirculation (on page 2, end of para 1), and added a statement on page 7 (para 3) pointing out that different vascular beds may have relative contributions of arterioles, capillaries and venules to the total vascular resistance.

Other

- How was It determined in fig 6 that the ratio of resistance between arterioles capillaries and venules is 1:4:2. Based on measured pressure drop along the arterial-capillary-venular tree I would expect the ratio to be more like 8:2:1.

The resistance ratios of 1:4:2 come from the paper of David Kleinfeld's group that we cite (Blinder et al., 2013) which used dye injection and 2-photon microscopy to document the topology of the cerebral vasculature, and use this morphology and the known variation of blood viscosity with vessel diameter to calculate the resistance of all the vessel segments. In that paper the resistance of the penetrating arteriole and ascending venule between the pia and layer 4 are stated explicitly as 0.1 and 0.2 P/micron³ at the bottom right of page 890, while the resistance between capillaries joining arterioles and venules in layer 4 was taken as the asymptotic network resistance in Fig 2g of the Blinder et al. paper, i.e. 0.4 P/micron³, thus giving the 1:4:2 ratios cited in the text. We have now stated this on page 7 (para 3). The difference with the referee's perspective comes from the fact that we are only considering the vessels within the brain parenchyma that may have their diameter regulated by the tissue in order to adjust its energy supply, which we have now stated explicitly on page 7 (para 3), while the referee appears to be considering all the vessels back to the aorta.

The ratio provided by the referee of 8:2:1 is cited as though it is derived from pressure drops measured in their previously cited reference (Chilian et al. Circulation Research vol-66 p-1227 1990) on the cardiac vasculature, which has a very different structure to that in the brain. However, that Chilian paper does not measure pressure drops, and we believe the referee may have meant to cite Chilian et al., 1986, Am J Physiol 251, H779). The latter paper measures pressures in large vessels to conclude that, in the vessel network from the aorta to the heart, 25% of the total resistance is upstream of 200 micron arterioles, while 20% is in arterioles from 100-200 microns in size, and the remaining 55% of the resistance is downstream of 100 micron arterioles. It is thus unclear where the stated 8:2:1 ratio comes from, but it appears technically difficult to use micropuncture measurements of pressure drops to estimate the resistance conferred by the capillary bed in any tissue.

We have now stated on page 7, para 3 that the 1:4:2 ratio used for the brain may differ significantly in other vascular beds.

- On pages 4-6 the argument is made that calculating F_{cap} (the ratio of a ratio) can determine the extent to which pericytes are playing a role in cerebral flow. It is not clear how the proposed means of changing pericyte activity would not also affect arterioles either directly or indirectly.

We have now inserted the following text on page 8 (para 1):

“Importantly, as considered in this section, there are conditions for which only the capillary resistance is altered, e.g. in Alzheimer’s disease model mice, pericytes are constricted while arterioles and venules do not change their diameter (Nortley et al., 2019, Fig. 5). Nevertheless, it will often be the case that arterioles and capillaries are dilated or constricted at the same time: we deal with this situation below.”

- On page 9 it is stated that in prior studies, responses in capillaries in 1-4 branch order are similar. How can that be if at the same time the density of pericytes decreases dramatically in larger vs. smaller branch order capillaries?

We were actually referring to the fact that the percent response rate of (visual cortical) pericytes to the increase of neuronal activity evoked by visual stimulation was not significantly affected by capillary branch order (Shaw et al., 2022, Table 1). There is a decrease of pericyte density with increasing branch order (Shaw et al., 2022, Fig. 7F), and the pericyte morphology also changes (having less circumferential processes at higher branch order), but those parameters are independent of response rate, as is the fact that the capillary dilation (measured as a percentage) evoked by neuronal activity is somewhat smaller for higher order capillary branches (Hall et al., 2014, Table 1). It is because of these variations in properties that we are careful to list the simplifying assumptions in our treatment on page 9 (para 2). These assumptions are necessary, in order to produce a simple and understandable model, which can be conveyed to the reader in a digestible manner in a brief Opinion piece.

We do attempt to demonstrate the effect of the increased spacing of pericytes on higher branch order capillaries in the following statement (page 8, para 4):

“The analysis above assumes that there are no spaces between pericytes where there is no constriction. This is a good assumption for pericytes on the first few branch orders of the capillary bed (where much of the resistance is located because for higher branch orders there are many more capillary segments in parallel) where it has been shown that the separation of pericyte somata is of the order of 20-40 μm (Grant et al., 2019; Shaw et al., 2022). However, on higher branch order capillaries, pericyte somata are further apart, perhaps $\sim 90 \mu\text{m}$ (Grant et al., 2019) and so if capillary constriction occurs for a distance up to 20 μm on either side of the soma (Nortley et al., 2019) there could be a 50 μm length of capillary in between the pericytes where no constriction occurs at all.”

which leads to Table 3 of the paper (in which we simplify the changes in pericyte density by grouping together higher and lower branch orders).

- It is stated that very small changes in diameter have dramatic effects on flow. This is correct, is explained in the Poiseuille equation, and applies to all vessels not just small ones. Post-arteriolar and arteriolar blood flow distribution should include a description of Poiseuille’s equation relating resistance as proportional to the change in radius 4. Therefore a 50% constriction yields a 94% reduction in flow.

We agree that Poiseuille’s law applies to all vessels, and indeed we have applied this law to arterioles as well as to capillaries in this work. Poiseuille’s law is shown in equation 2, and the effect of arteriolar constriction is analysed on page 10, para 3, and shown in Table 4. In addition to the effects of Poiseuille’s law, we explain in detail in the text and equations that

the blood viscosity increases at small vessel diameters, and this will also increase the resistance. The viscosity correction is particularly large for small capillaries, and less significant for arterioles, as explained in the text. Thus, it is not just Poiseuille's law but also the diameter dependence of viscosity that makes small diameter changes have a large effect in capillaries. This has now been highlighted in the text at the point where we first mention the large effect of small diameter changes (on page 5, point I), where we state that this is:

"as a result of both Poiseuille's law and the dependence of blood viscosity on diameter".

In relation to Fig. 4 the data shown reveal a much more attenuated response likely due to autoregulation or other compensatory mechanisms.

Figure 4 is a schematic representation of the effects of different degrees of hypoxia on the brain, summarising the literature. No data are shown.

Perhaps the referee means Table 4, but there can be no autoregulation or other compensatory mechanisms in Table 4, as this presents simply calculations based on Poiseuille's law (with viscosity changes accounted for), as outlined in the text and equations, parametrised in terms of the percentage change of vessel diameter that occurs.

Page 10: What is the teleological explanation for ischemic activation of vessel obstruction by circulating neutrophils?

Although we do not want to speculate on this in the text of our Opinion piece, one positive feature of neutrophils getting stuck at pericyte contraction points may be to promote the extravasation of these cells into the brain in pathology. Whether any beneficial aspect of their function once within the parenchyma outweighs the decrease of blood flow that they produce while stuck is a moot point.

Referee #2:

The paper entitled: A tight squeeze: how do we make sense of small changes in microvascular diameter? summarizes a point of view regarding the role for pericytes on brain capillaries for control of the cerebral circulation. The pericyte topic and the importance of distinguishing between arterioles and capillaries was recently reviewed in an excellent paper in The Journal of Physiology by Mughal et al. and this paper follows in the footsteps of that paper.

As noted above, the Mughal et al. review was published the day after we submitted our invited Opinion piece, and so the content of our piece was conceived independently of that work. We agree that Mughal et al. (2023) is a timely review on cerebral blood flow regulation, and we have now added a reference to it on page 5 Section (I) of our paper. However, while that review focusses on the post-arteriole transition zone (the first few capillary branches off of a penetrating arteriole), the topic of our Opinion piece is completely different, i.e. how to estimate the effect on vascular resistance of small changes in capillary diameter generated by pericytes. Thus, our commentary is complementary to other reviews such as that of Mughal et al. (2023).

For about a decade it has been documented that pericytes contribute to an important part of cerebrovascular resistance and this paper adds to the existing evidence using a modeling approach. The model is mainly based on excellent results obtained in the

Attwell lab, mainly in slices but also in vivo and the model faithfully reproduces the observations that the Attwell group have reported in several publications.

We thank the referee for their kind comments.

The model is simple and is based on the notion that pericytes constrict the underlying capillary close to a pericyte soma and that the diameter increases linearly between cell bodies. It is unclear whether this assumption is canonical in the pericyte/capillary field and to what extent this has been documented. This may be important because that notion is the basis for the extended version of the elegant model that is proposed by the authors.

We have established that, at least in the CNS and kidney, pericyte-mediated constriction of the underlying capillary is greatest at the pericyte somata and decreases approximately linearly with distance. Whilst we are not aware of any controversy regarding this (and it is certainly possible that the data might equally well be fitted by an exponential decrease of contractile influence with distance from the pericyte soma), we accept that it is important to direct the reader towards the evidence in order for them to form their own opinion on the matter. As such our text contained the line:

“Experimental data suggest that, in the presence of a vasoconstrictor, the diameter has a near linear relationship with distance from the cell soma (Nortley et al., 2019), as shown in Fig. 2.” to which we have now added the exact figure panels containing the data and have added a reference to similar data from kidney capillary pericytes (see page 6, para 2)

A second issue is that the description of the vascular network and the model do not include variations of the influence of pericytes across the microvascular bed, e.g., the very dynamic precapillary sphincters at the inlet and the very slow reacting pericytes at the venous end. The text also does not relate to previously published models of regulation of brain capillaries based on simple hemodynamic network models nor complex models of transit times. Still, this text is a valuable contribution to the pericyte literature giving novel insight to cerebrovascular resistance exerted by the capillary network in health and even more so in disease states.

In fact we explicitly considered the variation of the contractile properties of pericytes across the vascular bed in a paragraph (now on page 3) which states:

“In central nervous system pericytes, perhaps the best described pericyte population, the quantity of α -SMA expression negatively correlates with capillary branching order (i.e. expression is lower in capillaries that are more branches downstream from the originating arteriole: Hall et al., 2014; Rungta et al., 2018; Alarcon Martinez et al., 2018; Hartmann et al., 2021). This aligns with observations that lower order pericytes (branch order 1 to 4) can elicit larger and faster lumen diameter changes than higher branch order pericytes (branch order 5 to 9) (Hall et al., 2014; Hartmann et al., 2021). Although it is often assumed that this relationship reflects differences of pericyte α -SMA protein expression alone, it is not understood whether there is a linear relationship between α -SMA expression and contractility in pericytes. Indeed, prior work has detected an ability for higher order pericytes to reduce lumen diameter, despite the difficulty of detecting high levels of α -SMA expression in these cells (Hartmann et al., 2021)“.

We also attempted to address the effects of variations of pericyte density along vessels (albeit in a simple manner) in Table 3, by examining the influence of the greater interpericyte distance in higher branch orders. This was also addressed in the text by the following paragraph (page 8, para 4):

“The analysis above assumes that there are no spaces between pericytes where there is no constriction. This is a good assumption for pericytes on the first few branch orders of the capillary bed (where much of the resistance is located because for higher branch orders there are many more capillary segments in parallel) where it has been shown that the separation of pericyte somata is of the order of 20-40 μm (Grant et al., 2019; Shaw et al., 2022). However, on higher branch order capillaries, pericyte somata are further apart, perhaps $\sim 90 \mu\text{m}$ (Grant et al., 2019) and so if capillary constriction occurs for a distance up to 20 μm on either side of the soma (Nortley et al., 2019) there could be a 50 μm length of capillary in between the pericytes where no constriction occurs at all.”

To reflect the introduction of the term precapillary sphincter by the Lauritzen groups, we have now added the following text on page 3, end of para 2:

“Furthermore, the Lauritzen group has also named a population of capillary pericytes situated at the start of first-order capillaries as capillary “sphincters”, due to their unique morphology, spatially localised constriction and privileged position at the junction between arterioles and the capillary bed (Grubb et al., 2020). However, these cells are functionally similar to other pericytes and, as contractile cells residing within the capillary bed, sphincters fit within the original definition of pericytes (Zimmerman, 1923). Further, despite their potential physiological importance, only 28% of first order capillaries were demonstrated to exhibit sphincter-like pericytes (Grubb et al., 2020), which are therefore a minor population. For this reason, the term pericyte is used here to refer to all contractile cells residing in the capillary bed.”

We already addressed the dynamics of pericyte regulation of CBF with a paragraph on timing, now on page 11 (2nd para), to which we have now added a mention of “sphincters”, so that it reads:

“The time course over which pericytes change capillary lumen diameter gives pericytes a key role in regulating rapid blood flow changes. First and second order capillary dilations (including those mediated by contractile cells on 1st order capillaries sometimes referred to as sphincters: Grubb et al., 2020) are significantly faster than in arterioles (Hall et al., 2014; Khennouf et al., 2018), implying that pericytes can respond first to physiological energy demands such as neuronal activity. This implies that first and second order pericytes have a somewhat privileged role in controlling local energy supply through dilation, which may confer a unique and important physiological role. In contrast, higher branch order pericytes may change the capillary diameter with a slower time course (Hartmann et al., 2021), although they will still contribute to an overall change in resistance as explained above.”

In addition to the changes described above, we have also corrected a typographical error in eqn. (5).

Dear Dr Davis,

Re: JP-OP-2023-284207R1 "A tight squeeze: how do we make sense of small changes in microvascular diameter?" by Harvey Davis and David Attwell

We are pleased to tell you that your paper has been accepted for publication in The Journal of Physiology.

Authors should note that it is too late at this point to offer corrections prior to proofing. The accepted version will be published online, ahead of the copy edited and typeset version being made available. Major corrections at proof stage, such as changes to figures, will be referred to the Editors for approval before they can be incorporated. Only minor changes, such as to style and consistency, should be made at proof stage. Changes that need to be made after proof stage will usually require a formal correction notice.

All queries at proof stage should be sent to: TJP@wiley.com

Yours sincerely,

Peter Kohl
Senior Editor
The Journal of Physiology

P.S. - You can help your research get the attention it deserves! Check out Wiley's free Promotion Guide for best-practice recommendations for promoting your work at www.wileyauthors.com/eeo/guide. You can learn more about Wiley Editing Services which offers professional video, design, and writing services to create shareable video abstracts, infographics, conference posters, lay summaries, and research news stories for your research at www.wileyauthors.com/eeo/promotion.

IMPORTANT NOTICE ABOUT OPEN ACCESS: To assist authors whose funding agencies mandate public access to published research findings sooner than 12 months after publication The Journal of Physiology allows authors to pay an Open Access (OA) fee to have their papers made freely available immediately on publication.

You can check if your funder or institution has a Wiley Open Access Account here: <https://authorservices.wiley.com/author-resources/Journal-Authors/licensing-and-open-access/open-access/author-compliance-tool.html>

EDITOR COMMENTS

Many thanks for you contribution to JP!

Much appreciated.

Peter